# Breast Cancer MCF-7 Cells Acquire Heterogeneity during Successive Co-Culture with Hematopoietic and Bone Marrow-Derived Mesenchymal Stem/Stromal Cells

**DOI:** 10.3390/cells11223553

**Published:** 2022-11-10

**Authors:** Ruoxiang Wang, Xudong Wang, Liyuan Yin, Lijuan Yin, Gina Chia-Yi Chu, Peizhen Hu, Yan Ou, Yi Zhang, Michael S. Lewis, Stephen J. Pandol

**Affiliations:** 1Department of Medicine, Cedars-Sinai Medical Center, Los Angeles, CA 90048, USA; 2Department of Biomedical Sciences, Cedars-Sinai Medical Center, Los Angeles, CA 90048, USA; 3Department of Pathology, VA Greater Los Angeles Healthcare System, Los Angeles, CA 90073, USA

**Keywords:** breast cancer, metastasis, cell–cell interaction, bystander cells, hematopoietic cells, bone marrow-derived mesenchymal stem/stromal cells, co-culture, heterogeneity, gene expression

## Abstract

During disease progression and bone metastasis, breast tumor cells interact with various types of bystander cells residing in the tumor microenvironment. Such interactions prompt tumor cell heterogeneity. We used successive co-culture as an experimental model to examine cancer–bystander cell interaction. RMCF7-2, a clone of the human breast cancer MCF-7 cells tagged with a red fluorescent protein, was tracked for morphologic, behavioral, and gene expression changes. Co-cultured with various types of hematopoietic cells, RMCF7-2 adopted stable changes to a rounded shape in suspension growth of red fluorescent cells, from which derivative clones displayed marked expressional changes of marker proteins, including reduced E-cadherin and estrogen receptor α, and loss of progesterone receptor. In a successive co-culture with bone marrow-derived mesenchymal stem/stromal cells, the red fluorescent clones in suspension growth changed once more, adopting an attachment growth, but in diversified shapes. Red fluorescent clones recovered from the second-round co-culture were heterogeneous in morphology, but retained the altered marker protein expression while displaying increased proliferation, migration, and xenograft tumor formation. Interaction with bystander cells caused permanent morphologic, growth behavioral, and gene expressional changes under successive co-culture, which is a powerful model for studying cancer cell heterogeneity during breast cancer progression and metastasis.

## 1. Introduction

Breast tumor cells diversify from parental cells at the primary site into a heterogeneous population as seen in metastatic tumors [1,2,3,4,5,6,7,8,9]. Heterogeneity progression is a dynamic process encompassing the entire clinical history of tumor progression and metastasis. Tumor cell heterogeneity is the driving force of therapeutic resistance and distant metastasis, as it provides endless derivative variant cells to ensure that some will always survive anti-tumor insults, while some will always bear migration and invasion advantages. Delineating the underlying mechanism to effectively block tumor cell heterogeneity progression should be a rational strategy for breast cancer treatment.

The mechanism of tumor cell heterogeneity seems multifaceted [4,7]. Genomic damage, genetic mutation, and epigenetic anomalies may all induce phenotypic changes among tumor cells. Cancer stem cell properties, cancer cell lineage plasticity, and cell fate reprogramming may result in derivative variant formation. Puzzlingly, heterogeneity progression is a complex process motivated by reciprocal interaction between cancer and bystander cells [5,9,10,11]. Various resident cell types of the tumor microenvironment have been implicated. For instance, cells of the hematopoietic lineage play driver roles in the process. The bone marrow hematopoietic stem cell niche is a suitable space for heterogeneous tumor cells to survive and colonize [12,13]. Tumor-infiltrating immune cells from both the lymphoid and myeloid lineages are often tropistic to heterogeneity progression and metastasis [14,15], while tumor-associated macrophage infiltration may be suggestive of worsened prognosis [16,17]. Another prominent bystander cell type, the mesenchymal stem/stromal cells (MSCs), may be attracted to the tumor microenvironment to become cancer-associated fibroblasts [18,19,20] to promote heterogeneity progression by affecting tumor cell survival, therapeutic resistance, angiogenesis, and immune escape. Most importantly, bone marrow-derived MSCs (BM-MSCs) modulate bone metastasis by facilitating heterogeneous tumor cell colonization and proliferation to macro-metastasis [21]. As cancer heterogeneity progression and metastasis involve interactions with multiple bystander resident cell types, a suitable experimental model is necessary for dissecting the mechanism of breast cancer progression and metastasis.

We investigated the mechanism of heterogeneity progression by modeling cancer–bystander cell interaction in experimental mice and through in vitro co-culture, since previous studies defined BM-MSCs and other organ-specific MSCs as obligatory participants in this process. Co-inoculation with BM-MSCs, for example, can force non-tumorigenic prostate cancer cells to form xenograft tumors, while derivative clones recovered from these tumors displayed marked genomic and behavioral heterogeneity [22,23]. Importantly, these clones became metastatic upon a second round of co-inoculation, leading to bone metastases. We used various co-culture models to examine the interaction between cancer and mesenchymal stromal cells [24,25,26,27,28,29,30]. Cancer cells recovered from co-culture had permanent genotypic and phenotypic changes, indicative of increased heterogeneity and acquired malignancy [24,28,29,30]. Recently, we determined that, during co-culture, cancer-bystander cell fusion could lead to tumor cell heterogeneity [28,29,31]. These results revealed that bystander cells in the tumor microenvironment play a determinant role in heterogeneity progression and that interaction with BM-MSCs is a direct way for cancer cells to acquire bone metastatic potential.

In the current study, we investigated whether hematopoietic cells (HCs), another important bystander cell population infiltrating the tumor microenvironment, could interact with cancer cells to promote heterogeneity progression. Human breast cancer MCF-7 cells tagged with a red fluorescence protein (RFP) were co-cultured first with HCs and then with BM-MSCs. Stable morphologic, behavioral, and protein expressional changes in the cancer cells corroborate the role of bystander cells in cancer heterogeneity progression.

## 2. Materials and Methods

### 2.1. Cell Lines and Cell Culture

The source of the human prostate cancer LNCaP and C4-2 cell lines in our laboratory has been previously reported [22,29]. Human breast cancer MCF-7 and MDA-MB-231 cell lines, and human pancreatic cancer BxPC-3 and MiaPaCa-2 cells were purchased from American Type Culture Collection (ATCC, Manassas, VA, USA), and were maintained in Dulbecco’s Minimum Essential Medium (DMEM, Life Technologies, Carlsbad, CA, USA), which was changed to RPMI 1640 when MCF-7 cells were used in co-culture. For HCs used in the study, the human promyelocytic leukemia HL-60, monocytic leukemia THP-1, T-cell leukemia Jurkat, and the murine EML hematopoietic stem cells were from ATCC. Human mast-cell leukemia HMC-1 cells were from Sigma-Aldrich (St. Louis, MO, USA). Sources of Burkitt’s lymphoma CA46, Daudi, Namalwa, Raji, and Ramos cells were reported [32]. Murine T-cell hybridoma A1.1 and B-cell lymphoma Wehi-231 cells were from Dr. YF Shi (American Red Cross, Rockville, MD, USA). All the HCs were cultured in RPMI 1640 medium (Life Technologies). All complete media contained 10% fetal bovine serum (Atlanta Biologicals, Lawrenceville, GA, USA), penicillin (100 U/mL), and streptomycin (100 μg/mL). To prepare for EML cells, mouse stem cell factor (SCF, Gemini Bio-Products, West Sacramento, CA, USA) was added at 50 ng/mL. The SCF was removed when EML cells were used in co-culture. The normal human BM-MSC line, hBM-MSC, from a female donor [33] was obtained from the Tulane Center for Gene Therapy (Tulane University, New Orleans, LA, USA) and cultured in α Minimum Essential Medium (αMEM, Life Technologies) containing 20% FBS, 2 mM L-glutamine, penicillin (100 U/mL), and streptomycin (100 μg/mL). The αMEM was changed to RPMI 1640 medium when hBM-MSCs were used in co-culture. In this study, all the cells were cultured in regular CytoOne culture ware (USA Scientific, Ocala, FL, USA), and all cell passaging was with a 1:5 replating ratio. All the cell culture was performed at 37 °C in humidified atmospheric air supplemented with 5% CO_2_.

### 2.2. Source of Primary Cells

This study used 4 peripheral blood samples from clinical breast cancer patients before surgical treatment at the Cedars-Sinai Medical Center (CSMC), with informed written consent. Another 3 samples were provided by healthy donors. The use of human samples for research was approved by the institutional review board (IRB numbers: Pro 00025217 and Pro 00030418). As we previously reported [34], blood samples in lavender-top Vacutainers (Becton, Dickinson and Company, Franklin Lakes, NJ, USA) were transported to the Biobank Service, where plasma from the samples was removed after centrifugation, leaving packed blood cells ranging from 1.5 mL to 2.5 mL for use in this study. In addition, this study included 3 female Balb/c mice (Taconic Biosciences, Rensselaer, NY, USA) as sources of blood, splenocytes, and bone marrow-derived HCs (mBM-HCs). The use of mice was approved by the Institutional Animal Care and Use Committee (IACUC number 2999). Anesthetized mice were subjected to intraperitoneal injection of heparin (2000 IU/kg), followed by blood collection (500 µL) from the left ventricle. The spleen and femurs from each animal were dissected and placed in RPMI 1640 medium.

### 2.3. Nucleated Cell Isolation

To isolate peripheral blood mononuclear cells (PBMCs), packed blood cells were first diluted one-fold with a balanced salt solution containing 0.01% glucose, 5 µM CaCl_2_, 9.8 µM MgCl_2_, 540 µM KCl, 126 mM NaCl, and 14.5 mM Tris, pH 7.6. Every 2 mL of diluted sample was then mixed with 8 mL of a sterile hemolysis buffer (150 mM NH_4_Cl, 15 mM Tris, pH7.4, and 0.1 mM EDTA), and incubated till complete hemolysis, usually within 5 min. Centrifugation at 300× *g* for 10 min was used to recover PBMCs, which were washed twice in phosphate buffered saline, resuspended in 2 mL of RPMI 1640 complete medium, and counted. Splenocytes were isolated by mincing the spleen dissections between frosted glass slides, and mBM-HCs were flushed out from the femurs. Before being used in co-culture, both preparations were treated with the hemolysis buffer to remove red blood cells, followed by pre-incubation in 10-cm culture dish for 4 h to remove adherent cells. A TC20 automatic cell counter (Bio-Rad Laboratories, Hercules, CA, USA) was used to estimate cell sizes and to determine viable cell numbers based on trypan blue exclusion.

### 2.4. RFP Tagging

MCF-7 cells were tagged with RFP through transfection with linearized pAsRed2 eukaryotic expression vector with our reported protocol [27,35]. G418 (300 μg/mL) was used to select stably transfected cells, which were subsequently cloned by limiting dilution. RMCF7-2, a representative clone stably emitting red fluorescence, was used in this study.

### 2.5. Co-Culture with Bystander Cells

To co-culture with HCs, which were all in suspension growth, RMCF7-2 cells were grown on 6-well plates to form a monolayer of 75% confluence in 2 mL of culture medium per well. HCs in another 2-mL of culture medium were overlaid to the monolayer, so RMCF7-2 and the HCs were in a 1:10 ratio in 4 mL of culture medium. The co-culture was kept for at least 2 weeks. To feed cells, 3 mL of the 4 mL medium was replaced gently with fresh medium every 3 days. Co-culture with hBM-MSCs was done in our reported cancer-stromal co-culture setting [27,28,29]. Briefly, hBM-MSCs were plated first to form a stromal cell monolayer on 6-well plates. Cells in suspension growth were overlaid to the monolayer in a 1:1 cell ratio. Control groups included co-cultures with dead HCs, which were HL-60 and A1.1 cells killed by fixation or snap-freezing, as we reported [29,36]. Conditioned media from 48 h cultures of HL-60 and A1.1 cells were also used as control.

### 2.6. Genotyping for Cell Lineage Authentication

We reported the use of short tandem repeat (STR) genotyping with the commercial Human STR Profiling Cell Authentication Service (ATCC) [28]. Cell lineage relationships were identified by searching through the ATCC genotype database.

### 2.7. Western Blotting

Our previously reported western blotting protocol was used [37,38]. Primary antibodies used in this study included those to E-cadherin (E-cad, 3195S) and progesterone receptor (PR, 3153S) from Cell Signaling Technology (Danvers, MA, USA). Antibodies to vimentin (Vim, sc-6260), estrogen receptor α (ERα, sc-8002), estrogen receptor β (ERβ, sc-390243), Her2/neu (sc-33684), epidermal growth factor receptor 1 (EGFR1, sc-373746), and β-actin (sc-69879) were from Santa Cruz Biotechnology, Inc. (Dallas, TX, USA). ER1D5, another antibody to human ERα [39], was from Immunotech Inc. (Westbrook, ME, USA). All the primary antibodies were used at 1 µg/mL working concentration.

### 2.8. Phenotypic Characterizations

To determine rate of proliferation, 2 × 10^5^ cells in 4 mL medium were seeded to each well of a 6-well plate. The cells were cultured for 6 days. Cells in triplicate wells were collected daily and counted with the TC20 counter under trypan blue exclusion conditions. To assess migration capacity, transwell assays were conducted on 24-well plates with inserts of 8 µm pore size (ThermoFisher Scientific, Carlsbad, CA, USA). For each cell sample, quadruplicate assays were performed. An equal number of cells (2 × 10^5^) in the upper chamber were cultured for 48 h. After the removal of inserts, cells migrated to the lower chamber were collected to the bottom by centrifugation at 300× *g* for 10 min. The cells were then quantitated with the method of crystal violet staining, as we reported [27]. To evaluate the potential of xenograft tumor formation, 6-week-old female NCr^nu/nu^ mice were used with our reported inoculation protocol [40]. For each group, 6 mice were inoculated in the fat pad of the fourth pair of mammary glands (*n* = 6). For each fat pad, 2 × 10^6^ cells were inoculated in 100 µL of 50% Matrigel (ThermoFisher Scientific). Tumor dimension was measured weekly with a caliper, and tumor volume was calculated as length × width^2^ × π/6. Tumor burden of 1.5 cm^3^ was used as a humane endpoint.

### 2.9. Fluorescence Microscopic Analysis

The protocol for fluorescence imaging was previously reported [28,29]. In this study, for comparison purposes, all red fluorescent images were taken with fixed settings: 24 s for imaging at 40× magnification, 8 s for imaging at 100× magnification, and 4 s for imaging at 200× magnification.

## 3. Results

We initially tested selected human prostate, breast, and pancreatic cancer cells for their interaction with various types of HCs. Subsequently, we investigated the consequence of the interaction with MCF-7 cells because these cells displayed a typical epithelial morphology with a well-defined marker protein profile, better facilitating the detection of heterogeneity progression.

### 3.1. RFP-Tagging to Track the Fate of MCF-7 Cells

Cells of the hematopoietic lineage form a significant component of bystander cells in the tumor microenvironment, as various types of HCs are tumor infiltrating cells and display tropism to metastasis. We conducted a series of co-cultures to assess the consequences of the interaction. MCF-7 cells in culture display a large epithelial cell shape and an attachment growth behavior with intercellular contact to form cobblestone-like arrangements, easily distinguishable from the small and round HCs in suspension growth (Figure 1A). We tagged MCF-7 cells with RFP expression and isolated the first six clones by limiting dilution. Through 30 continued passages, we determined that these clones were stable, retaining the morphology and growth behavior of the parental MCF-7 cells while displaying uniform RFP expression. A representative clone, RMCF7-2, was randomly selected and used in this study (Figure 1B). RFP-expression could be used as a reliable marker for tracking the fate of cancer cells in co-culture [27,28,29,30].

### 3.2. Epithelial RMCF7-2 Cells Adopting Suspension Growth upon Co-Culture with HCs

We conducted co-culture by overlaying different types of HCs on an RMCF7-2 cell monolayer (Figure 1B). HCs tested included PBMCs derived from healthy donors and breast cancer patients; and established human cell lines. Mouse PBMCs, splenocytes, and mBM-HCs, together with murine cell lines, were also used. Cells in co-culture were inspected daily for 2 weeks, and individual co-cultures were repeated at least three times. Co-culture with different HCs all resulted in morphologic and behavioral changes in the RMCF7-2 cells. In the co-culture with a breast cancer patient’s PBMCs, for example, individual RMCF7-2 cells in the monolayer would shrink to lose contact with each other in the first week, concomitant with the appearance of large and round red fluorescent cells in suspension. Besides the fluorescence emission, these cells were of MCF7-2 origin because they were around 15 µm in diameter as determined by the TC20 autocounter, much larger than the adjacent PBMCs (~10 µm), but close to the size of RMCF7-2 cells when they were trypsinized and prepared in single-cell suspension. RMCF7-2 cells in attachment became fewer in the second week, as most became detached to adopt suspension growth (Figure 2). By the end of a 2-week co-culture, there were virtually no cells in the attached growth, while red fluorescent cells in suspension adopted a loosely clustered growth that was quite different from mammospheres in which MCF-7 cells form tightly packed 3-dimensional spheroid growth, as we evaluated in a separate study [40].

The loss of intercellular contact to assume suspension growth seemed to be an innate or programmed response, because the same was seen in RMCF7-2 cells in co-culture with all 11 types of human and six types of mouse HCs (Table 1). Importantly, the same changes were seen from co-cultures with all primary PBMC preparations from three healthy donors and four breast cancer patients, as well as samples of PBMCs, splenocytes, and mBM-HCs from mice (Figure 3). We previously established that, when cultured alone, most PBMCs would die in 2 weeks under the specified conditions of cancer cell culture [41]. Whether certain PBMCs in these co-cultures blasted or proliferated remains to be determined. Regardless, this series of co-cultures demonstrated that, no matter which type of HCs were used, RMCF7-2 cells would respond with similar morphologic and behavioral changes. In parallel control studies, we determined that a 2-week culture in 50% HC-conditioned media, or co-culture with dead HL-60 or A1.1 cells could not convert RMCF7-2 cells to suspension growth. Thus, it was the viable HCs in direct contact that induced RMCF7-2 cells to change morphology and growth behavior.

### 3.3. Stability of the Morphologic and Behavioral Changes

To assess whether the observed changes were transient or a permanent switch, we purged any remaining HCs by treating the co-culture with G418 (300 µg/mL) for another 2 weeks, leaving red fluorescent cells in suspension to be subjected to continued culture. In this series of cultures, red fluorescent cells from co-cultures remained proliferative beyond passage 15, suggesting that the acquired suspension growth was a stable phenotype.

To establish clones for detailed analysis, we subjected the G418-treated red fluorescent cells at passage 15 to limiting dilution cloning by diluting 120 cells to 384 wells. These cells displayed robust colony formation, as more than 30 clones were formed from individual limiting dilution. The results of the cloning studies are summarized in Table 1.

We performed continued passaging to further assess the cloned cells for their stability in suspension growth. Two clones, MAF-2 and MAF-12 (for MCF-7 and floating, Figure 4A), were randomly selected from the 12 clones derived from the co-culture of RMCF7-2 with PBMCs of the second breast cancer patient. We used STR genotyping to confirm that the red fluorescent clones in suspension growth were indeed of the MCF-7/RMCF7-2 cell lineage, as MAF-2 and MAF-12 were found to carry a genomic fingerprint identical to both the parental RMCF7-2 and grand-parental MCF-7 cells (Figure 4B). Though these derivative clones in suspension were generally slower in growth rate than the parental RMCF7-2 cells, the two clones were cultured to 60 passages without signs of losing suspension growth. The acquired features of HC-like morphology and suspension growth, therefore, were quite stable.

In parallel studies, other cancer cells tested (i.e., LNCaP, C4-2, MBA-MD-231, BX-PC-3, and MiaPaCa-2) displayed similar morphologic and growth behavioral changes, becoming rounded and losing attachment to adopt suspension growth upon co-culture with HCs. Whether these cells would adopt a stable behavior of suspension growth up to 60 passages remains to be determined.

### 3.4. Co-Culture-Induced Marker Protein Expression Changes

To characterize gene expressional changes accompanying the morphology and behavior switch, we examined the isolated derivative clones by focusing on well-defined marker proteins for MCF-7 cells. Substantial expressional changes were identified by western blotting. The MAF-2 and MAF-12 clones at passage 30 expressed markedly reduced ERα as detected by the sc-8002 antibody. When the ER1D5 antibody was used, ERα in these clones was seen as two faster migrating bands suggestive of protein cleavage (Figure 4C). Another prominent change was the loss of PR expression. In comparison, little change was seen in the ERβ or Her2/neu level, although the EGFR1 level was elevated. Intriguingly, as these cells had given up the epithelial morphology, substantial E-cad persisted, albeit at lowered level, while vimentin was not detected in these cells. The expressional changes persisted from passages 30 to 60, because similar expression patterns were seen in these clones beyond p60. Though comprehensive expressional profiling has yet to be conducted on all the available clones (Table 1), our results demonstrate that the stable phenotypic switch is accompanied by stable changes of gene expression.

### 3.5. Resuming Attachment Growth through Co-Culture with hBM-MSCs

Breast cancer frequently metastasizes to the bone. During this process, metastasizing cells may experience successive interactions with different bystander cells, including BM-MSCs [5,9,10,11]. We conducted another round of co-culture by overlaying cloned MAF-2 and MAF-12 cells to a monolayer of female donor hBM-MSCs. Cell morphology and growth behavior in the co-culture were monitored daily for 4 weeks.

Microscopic inspection of the second-round co-culture revealed that the MAF-2 and MAF-12 cells in suspension growth resumed attachment growth in the presence of the stromal monolayer. Many MAF-2 and MAF-12 cells, for instance, attached to hBM-MSCs in the first week, with individual cells giving up the round shape to adopt attachment growth in the second week. There were frequent cell fusion events between cancer and hBM-MSCs (Figure 5A), seen as red fluorescent cells but with hBM-MSC morphology, as we previously reported [28,29]. In the third week, red fluorescent cells started appearing in groups, most probably because of clonal expansion (Figure 5B). Intriguingly, cell morphology of the newly formed attachment growth was divergent not only from the parental MAF-2 or MAF-12 cells but also from the grand-parental RMCF7-2 cells, as no typical epithelial shape or cobblestone-like arrangement was ever observed. Instead, these cells were seen in short spindle shapes without intercellular contact, in scattered arrangements, or with overlapping growth, while some cells in attachment growth still kept a rounded cell shape (Figure 5A,B). By the end of a 4-week co-culture, the hBM-MSC monolayer was teeming with heterogeneous cancer cell clusters, representing individual red fluorescent clones of distinct clonal morphology and growth behavior. These results once more demonstrated that interaction with bystander cells changed cancer cell morphology and growth behavior. Because cancer–MSC interaction takes place between individual cells, the interaction may affect cancer cells individually, causing inter-clonal heterogeneity.

### 3.6. Stability of Red Fluorescent Cells after Resuming Attachment Growth

To isolate red fluorescent cancer cells that resumed attachment growth from a 4-week co-culture with hBM-MSCs, G418 (300 µg/mL) was added for 2 weeks to purge the stromal cells. Cells of representative red fluorescent colonies were picked up with cloning disks and cloned by limiting dilution, and the first five clones were kept for passaging. For this report, the morphology of the five clones (MAF-BMSC-1 to -5) derived from the MAF-2 co-culture with hBM-MSCs is shown (Figure 6A). The distinct morphologies and resumed attachment growth behavior were stable, as these clones kept the same features up to 60 passages.

We conducted another round of western blotting to assess the status of marker protein expression. Despite re-adopted attachment growth, all five MAF-BMSC clones maintained an expressional pattern rather closer to the parental MAF-2 cells than the grand-parental RMCF7-2 cells, showing reduced ERα expression, loss of PR expression, and elevated EGFR1 level (Figure 6B). Though these clones had resumed attachment growth, E-cad levels remained low. Vimentin remained undetectable. Clones isolated from the second round of co-culture were thus discretely heterogeneous from both the parental MAF-2 and the grand-parental RMCF7-2 cells.

### 3.7. Acquired Aggressive Behaviors from Successive Co-Culture

We assessed whether successive co-culture results in altered cancer cells with behavioral heterogeneity from parental cells. Compared to RMCF7-2 cells, the clones chosen from the successive co-culture (MAF-BMSC-1, -3, and -5) all showed faster cell proliferation (Figure 7A), higher migratory capacity (Figure 7B), and larger xenograft tumor formation (Figure 7C), although the behavioral changes varied in degrees among the clones. Intriguingly, both MAF-2 and MAF-12 were slow in proliferation and migration (Figure 7A,B), while repeated animal studies with the same inoculation protocol indicated that MAF-2 and MAF-12 cells could not form xenograft tumors (Figure 7C). Further examination is needed to determine whether other clones from RMCF7-2 and HC co-culture also lost xenograft tumor-formation potential.

## 4. Discussion

The process of breast cancer progression and metastasis is substantiated by tumor cell heterogeneity, which is in a dynamic state most probably sustained by continuous interactions between cancer and bystander cells [5,9,10,11]. Illustrating the mechanism of tumor cell heterogeneity progression is the goal of cancer metastasis research.

This study used successive co-cultures to simulate cancer cell interaction with infiltrating HCs and the resident BM-MSCs in the bone tumor microenvironment. A representative clone of RFP-tagged MCF-7 cells, RMCF7-2 (Figure 1), was tracked for cellular heterogeneity progression. Through co-cultures with various types of HCs, RMCF7-2 cells underwent morphologic changes and adopted a suspension growth (Figure 1, Figure 2 and Figure 3 and Table 1). The change was not due to cloning deviation, because our unpresented data suggest that the MCF-7 cells per se could be converted to suspension growth as well. The morphologic and behavioral changes were accompanied by gene expressional changes, as the derivative MAF-2 and MAF-12 clones (Figure 4A,B) were detected with altered expressions of the E-cad, ERα, PR, and EGFR1 proteins (Figure 4C). By the second round of co-culture, the MAF-2 and MAF-12 cells in suspension growth could switch again, adopting attached growth in the presence of hBM-MSCs (Figure 5A) to form heterogeneous clones with mutually diversified morphologies and growth behaviors (Figure 5B). The marker protein expression pattern in the first five derivative clones was more similar to the parental MAF-2 cells than to the grand-parental RMCF7-2 cells. Thus, successive co-culture with two unique bystander cell types rendered derivative cells that had diversified from both the parental and grand-parental cancer cells. The diversification affected cancer cell aggressiveness, as clones derived from second-round co-culture were found with an increased proliferation rate (Figure 7A) and migration capacity (Figure 7B), together with accelerated tumor formation (Figure 7C). Significantly, our unpresented data documented similar morphologic and behavioral changes in the co-culture between HCs and other human cancer cells of the prostate, breast, and pancreas. Interaction with infiltrating HCs in the tumor microenvironment may be a shared strategy for cancer cell heterogeneity progression.

Related to this study, additional behavioral analyses are needed with other MAF clones to determine whether the co-culture between RMCF7-2 and HCs rendered all the cancer cells in suspension less aggressive as measured with the same parameters (Figure 7). Analyses for additional gene expressional changes are also needed to unveil other accompanying expressional changes with the newly acquired aggressiveness following the second-round co-culture. Notwithstanding, several significant findings from this study are worthy of discussion.

### 4.1. Interaction with HCs as a Mechanism of Circulating Tumor Cell (CTC) Formation

It is well established that MCF-7 cells will adapt to agglomerate or spheroid growth in non-adherent or ultra-low attachment culture wares [42,43], on specified substrates, or in defined medium containing growth factors of epidermal growth factor, basic fibroblast growth factor, or insulin or insulin-like growth factor 1 [44]. The adaptation is, however, conditional to the specified treatments, since MCF-7 cells resume attachment growth once placed back to normal culture conditions. In this study, the switch to stable suspension growth took place in regular cell culture wares, while HC-conditioned media could not convert RMCF7-2 cells to suspension growth, suggesting the importance of direct interaction with viable HCs. This finding may have a strong clinical relevance because breast tumor is frequently infiltrated by various types of HCs [14,15,16,17]. In this study, whether the cancer cells in suspension growth, such as the MAF-2 and MAF-12 clones, represent bona fide CTCs remains to be characterized. Should clinical breast tumor cells adopt similar morphologic and growth behavioral changes in situ through interaction with infiltrating HCs, CTCs could be formed for systemic spreading. It would be intriguing to investigate whether cancer cell interaction with infiltrating HCs in the tumor microenvironment is the cause of CTC formation.

### 4.2. Phenotypic Changes Caused by Interaction with HCs and Lineage Plasticity in Cancer Cells

Through successive co-culture, RMCF7-2 cells underwent two marked morphology switches, from a typical epithelial to a HC-like morphology (Table 1, and Figure 1, Figure 2, Figure 3 and Figure 4), and then to heterogeneous epithelial shapes again (Figure 5 and Figure 6). Should the observed changes manifest lineage plasticity, our study would suggest that lineage plasticity in cancer cells may be triggered externally by interaction with bystander cells. It is imperative to elucidate the mechanism by which RMCF7-2 cells adopt morphologic and growth behavioral changes in interacting with different bystander cells.

### 4.3. Interaction with HCs as a Model of Malignant Progression

The marker gene expression changes caused by co-culture were another intriguing finding. Whereas various genomic, genetic, and epigenetic mechanisms inside a cancer cell could account for gene expression changes, results from this study strongly suggested that the loss of marker protein expression is initiated by cancer–bystander cell interaction. With specific antibodies and western blotting, we defined MCF-7 and its RMCF7-2 clone in this study with an ER^+^PR^+^Her2/neu^low^EGFR1^low^ status (Figure 4C). RMCF7-2 co-culture with PBMCs led to derivative cancer cells with irreversible ERα suppression and PR loss, while EGFR1 levels were elevated, especially after the second co-culture with hBM-MSCs (Figure 6B). ERα suppression, PR loss, and EGFR1 elevation are signs of a worse prognosis, as breast tumors` with these features may have more malignant potential [45,46]. To fully appreciate gene expressional changes and possible expressional heterogeneity, a comprehensive expressional analysis needs to be conducted. In this study, we isolated several groups of clones (Table 1) for detailed genetic and gene expression analyses. A comprehensive cellular characterization and xenograft tumor formation remains to be conducted to assess for increased tumorigenicity in these clones.

### 4.4. The Mechanism of Cancer Cell Changes Caused by Interaction with HCs

Currently, it is not clear how cancer cells may acquire such significant morphologic, behavioral, and gene expressional changes simply through co-culture with bystander HCs. In isolated clones from the co-cultures, the high stability of the changes strongly suggested a shift of genomic, genetic, or epigenetic control. In this regard, reciprocal humoral communication is thought to be the major route of co-evolution of cancer and bystander cells. Growth factors, cytokines, chemokines, metabolic intermediates, extracellular matrices, and extracellular vesicles enclosing genomic materials and macromolecules could all function as mediators in cell–cell interaction [47,48]. In this study, however, co-culture with dead HCs or treatment with HC-conditioned media did not cause discernible RMCF7-2 changes, suggesting a critical role of direct interaction with live HCs. Elucidating how the interaction switches the control machinery in cancer cells will be critical to understanding the process of cancer progression and metastasis.

Besides soluble factor-mediated intercellular communication, breast cancer bone metastasis involves interactions with bystander cells in the bone. We recovered cancer cell clones, MAF-2 and MAF-12, from co-culture with HCs and assessed their fate upon interaction with the hBM-MSCs. In these co-cultures, frequent cell fusion between MAF-2 or MAF-12 cells and the hBM-MSC monolayer was observed. Cancer cell fusion with bystander cells is a direct route to tumor cell heterogeneity [28,29,31]. It is possible that the observed morphologic and growth behavioral changes in successive co-culture were due to cancer–bystander cell fusion, as certain fusion hybrids may survive and proliferate to form heterogeneous progenies, each carrying certain traits of the parental cells. On the other hand, STR genotyping did not find signs of genomic hybridization in MAF-2 or MAF-12 clones (Figure 4B). Recently, we determined that cancer–bystander cell fusion could be studied by tagging cancer and bystander cells with distinct fluorescence proteins [28,29]. This strategy remains to be tested to assess the role of cancer–bystander cell fusion in modulating the changes in RMCF7-2 cells during successive co-cultures.

### 4.5. Limitations of the Study

Using RFP as a cancer cell marker, we can track heterogeneity progression and clone the changed cells from a complex co-culture system. These clones may serve as subjects to gauge the extensiveness of the heterogeneity, which becomes exponential in consecutive co-culture (Figure 5 and Figure 6). Heterogeneity progression may be accompanied by altered expressions of many proteins. While their action role remains to be elucidated, regulatory proteins may modulate the process of disease progression. We found that stemness-related proteins may promote tumorigenicity in MCF-7 cells [40], while many transcription factors, including androgen receptor [49], are active in this cell line. This study detected several well-characterized proteins in MCF-7 cells to demonstrate that interaction with HCs could induce expressional changes. With a reproducible protocol to isolate derivative subclones, transcriptomic and proteomic methods will be used to assess the scope of altered expression.

This study took six widely used cell lines of the human breast, prostate, and pancreatic cancers as experimental subjects. Like MCF-7 cells, the other cell lines all responded to the co-culture with HCs by changing to suspension growth. Whether these changed cells could sustain the growth behavior in continued culture remains to be tested, and whether interaction with HCs in vivo may convert tumor cells with similar changes has not been corroborated. In this regard, we have observed progressive and permanent morphologic and gene expressional changes during successive xenografting [35], where prostate cancer cells with epithelial shapes would adopt a CTC-like morphology to be disseminated from the orthotopic site. Culture of blood samples from the host mice yielded altered cancer cells with drastic mesenchymal stromal morphology and protein expression.

## Figures and Tables

**Figure 1 cells-11-03553-f001:**
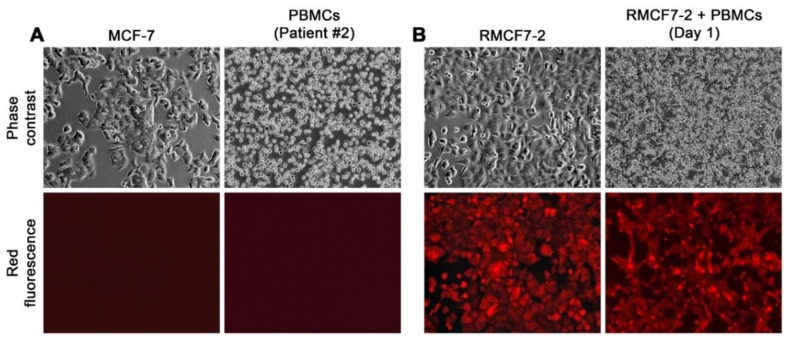
Using RFP tagging to facilitate cell tracking in co-culture. A representative co-culture is used to demonstrate the differences of MCF-7 from HCs. All the images are shown at 100× magnification. (**A**) Marked differences in morphology and growth behavior between MCF-7 and PBMCs derived from the second breast cancer patient (Patient #2). (**B**) A co-culture setup on the first day (Day 1), with an RMCF7-2 cell monolayer overlaid by PBMCs of the Patient #2. RMCF7-2 cells in the co-culture could be tracked with red fluorescence microscopic imaging.

**Figure 2 cells-11-03553-f002:**
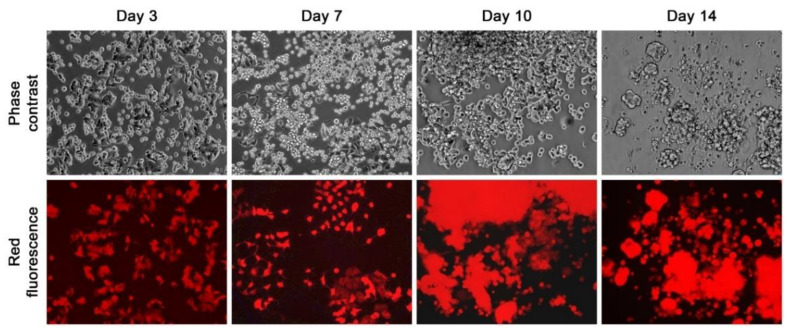
RMCF7-2 morphology and growth behavior during co-culture with HCs. A representative co-culture with Patient #2’s PBMCs demonstrates the gradual changes in the red fluorescent RMCF7-2 cells from Day 3 to Day 14. All the images are shown at 100× magnification.

**Figure 3 cells-11-03553-f003:**
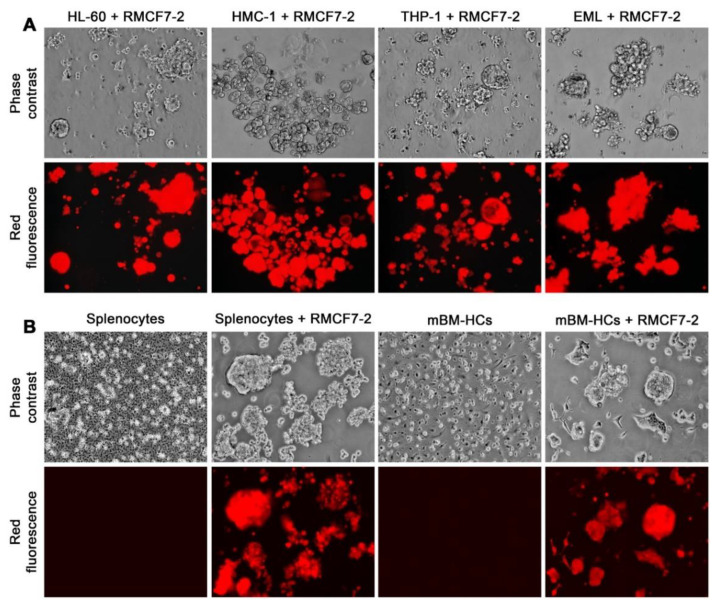
Suspension growth as an innate or programmed response. Similar morphology and growth behavior changes in RMCF7-2 cells after 28 days of co-culture with different types of HCs. Representative results are shown at 100× magnification. (**A**) Similar changes in co-cultures with varying lines of HCs. In these studies, HCs in the co-culture were killed by a 2-week G418 treatment. (**B**) Similar changes in co-cultures with murine splenocytes or mBM-HCs for 14 days. In these studies, mono-cultures of splenocytes and mBM-HCs were used as control. HCs in these co-cultures were not removed by G418 treatment.

**Figure 4 cells-11-03553-f004:**
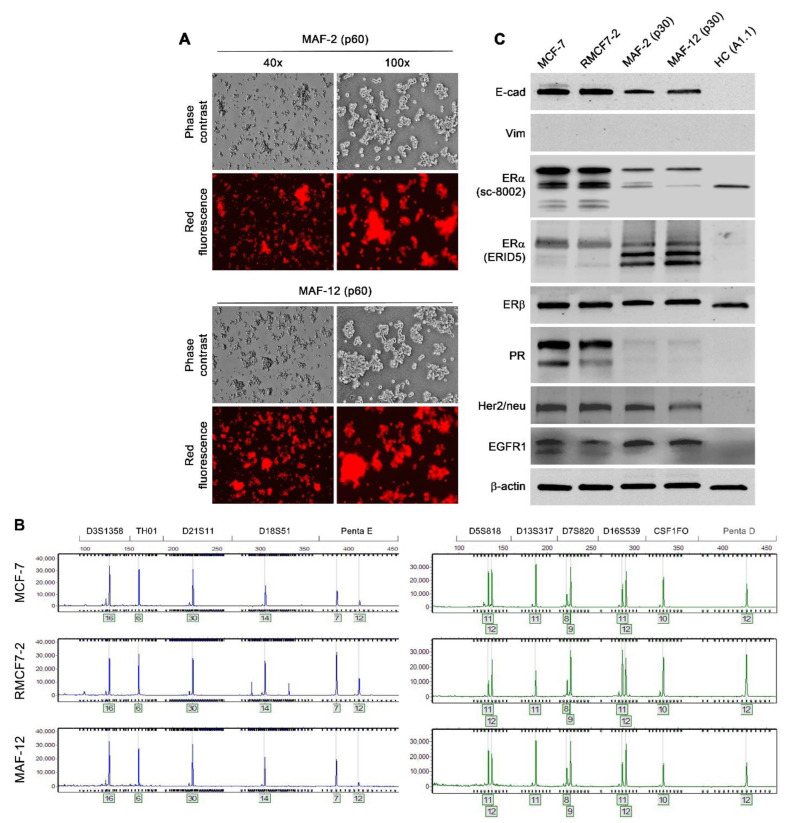
Stability of morphologic, growth behavioral and gene expressional changes. Representative results are shown. (**A**) MAF-2 and MAF-12 clones isolated from RMCF7-2 co-culture with breast cancer patient PBMCs were subjected to continuous culture in a 1:5 replating ratio. Cell images were taken at passage 60 (p60). The HC-like morphology and suspension growth behavior described in Figure 2 were well preserved throughout the passages. (**B**) Genotypic identity of MAF-12 to RMCF7-2 and MCF-7 cells. Cells of the MAF-2 clone carried the same STR profiles. Including the results not shown, all the 18 STR loci analyzed were identical among these cells. (**C**) Altered marker protein expression in MAF-2 and MAF-12 clones at p30. The expressional changes were stable since similar results were obtained with MAF-2 and MAF-12 clones at p60. For comparison, murine T cell hybridoma A1.1 cells were used as an HC control.

**Figure 5 cells-11-03553-f005:**
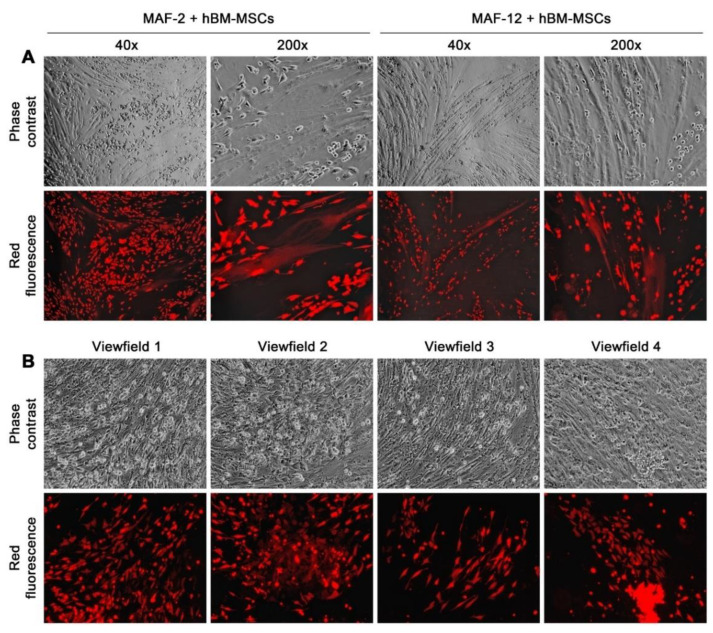
Second round co-culture with hBM-MSCs restored MAF-2 and MAF-12 cells to an altered attachment growth behavior. (**A**) After co-culture with hBM-MSCs for 14 days, many MAF-2 and MAF-12 cells adapted attachment growth. Some were involved in fusion with cells of the hBM-MSC monolayer, seen as large hBM-MSCs with red fluorescence, which is easier to identify at 200× magnification. (**B**) Images of four low-magnification (40×) view fields show that, as the co-cultures proceeded beyond 3 weeks, many red fluorescent clones could be seen in the hBM-MSC monolayer, with individual clones displaying distinct cell morphology and growth behavior.

**Figure 6 cells-11-03553-f006:**
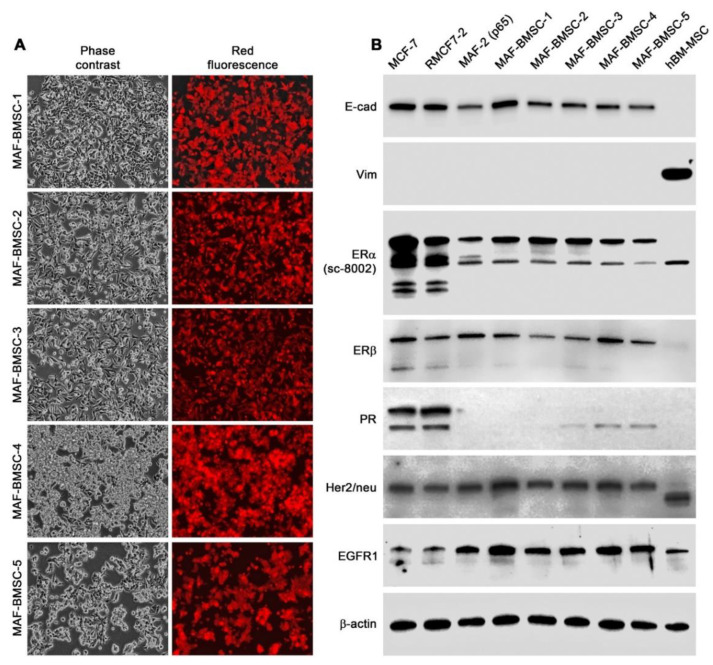
Morphology and marker protein expression of clones isolated from the second round of co-culture. Characteristics of the first five MAF-BMSC-clones are shown. (**A**) Diversified morphology and growth behavior of the clones at passage 30 (40×). (**B**) The clones at p60 still maintained the pattern of marker expression of the parental MAF-2 clone (compared to Figure 4C).

**Figure 7 cells-11-03553-f007:**
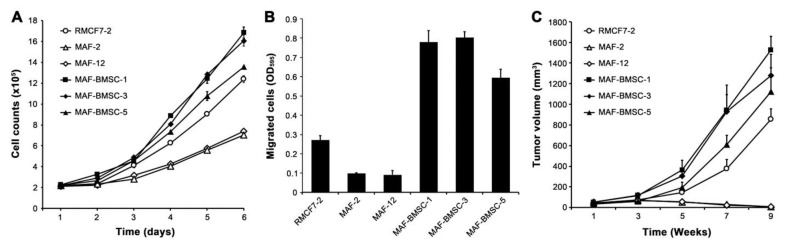
Co-culture with bystander cells induced behavioral heterogeneity. Selected MAF-BMCS-clones (-1, -3, and -5) were compared with the parental MAF clones (-2 and -12) and the grand-parental RMCF7-2 clone for behavioral changes. (**A**) Growth rates were compared by counting the growth of an equal number of cells daily. Data are shown as the cell count average of triplicate wells ± STDEV. (**B**) Migration capabilities were compared by seeding an equal number of cells in a 48-h transwell assay. Data are shown as the crystal violet stain average of triplicate wells ± STDEV. (**C**) Increased xenograft tumor formation capability in MAF-BMSC-clones was revealed through orthotopic mammary fat pad tumor formation (*n* = 6). Null of tumor formation by MAF-2 and MAF-12 cells was confirmed with additional animal studies. In all the assays, data are shown as tumor volume average ± STDEV.

**Table 1 cells-11-03553-t001:** Appearance of red florescent cells in suspension growth from co-cultures with RMCF7-2 cells.

Samples Co-Cultured with RMCF7-2 Cells	Suspension Growth Observed?	Cloned by Limiting Dilution?
Healthy donor PBMCs (3 donors)	Yes, with all 3 samples	No
Clinical breast cancer patient PBMCs (4 patients)	Yes, with all 4 samples	Yes, the first 12 MAF clones (including MAF-2 and MAF-12) from the second sample
HL-60, human promyelocytic leukemia cell line	Yes, in 3 co-cultures	Yes, the first 24 clones
HMC-1, human mast cell leukemia cell line	Yes, in 3 co-cultures	Yes, the first w clones
THP-1, human monocytic leukemia cell line	Yes, in 3 co-cultures	Yes, the first 6 clones
Jurkat, human T cell leukemia cell line	Yes, in 3 co-cultures	No
CA46, human Burkitt’s lymphoma cell line	Yes, in 3 co-cultures	No
Daudi, human Burkitt’s lymphoma cell line	Yes, in 3 co-cultures	No
Namalwa, human Burkitt’s lymphoma cell line	Yes, in 3 co-cultures	No
Raji, human Burkitt’s lymphoma cell line	Yes, in 3 co-cultures	No
Ramos, human Burkitt’s lymphoma cell line	Yes, in 3 co-cultures	No
Balb/c mouse PBMCs (3 mice)	Yes, with all 3 mice	No
Balb/c mouse spleen cells (2 mice)	Yes, with the 2 mice tested	No
Balb/c mBM-HCs (2 mice)	Yes, with the 2 mice tested	Yes, the first 2 clones from 1 mouse
A1.1, mouse T cell hybridoma cell line	Yes, in 6 co-cultures	Yes, the first 6 clones
EML, mouse multipotent hematopoietic cell line	Yes, in 3 co-cultures	Yes, the first 6 clones
>WEHI-231, mouse B cell lymphoma cell line	Yes, in 3 co-cultures	No

## Data Availability

Representative data for this study are included in this published article. Contact the corresponding author for additional data.

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
