# Peer review of "Breast Cancer MCF-7 Cells Acquire Heterogeneity during Successive Co-Culture with Hematopoietic and Bone Marrow-Derived Mesenchymal Stem/Stromal Cells"

_cells, 2022, doi:10.3390/cells11223553_

Round 1
Reviewer 1 Report
Authors conducted a good work. However, in my opinion should be summarized and some information could be shifted in supplementary materials. In addition, considering the relevant impact played by androgen receptor in breast cancer and the pivotal role of cancer stem cells, western blot analysis should be conducted also to assess the AR expression and stem cell markers. To deepen the background I strongly suggest the following manuscripts (doi: 10.4252/wjsc.v11.i9.594; https://doi.org/10.3389/fendo.2018.0049).
Author Response
- However, in my opinion should be summarized and some information could be shifted in supplementary materials.
Our response: We thank the Reviewer for the suggestion. During the revision process, we realized that all the presented data were integral to the main theme of the manuscript, which describes the natural history of the cancer cell interaction with hematopoietic cells, from the beginning of the interaction to the consequent heterogeneity progression. Removal of any part of the data may compromise continuity of the story telling and introduce unnecessary confusion to the readers.
- In addition, considering the relevant impact played by androgen receptor in breast cancer and the pivotal role of cancer stem cells, western blot analysis should be conducted also to assess the AR expression and stem cell markers.
Our response: We thank for the Reviewer’s suggestion, which insightfully points to the potential control mechanism of heterogeneity progression. As an initial report of our ongoing research on cancer cell interaction with hematopoietic cells, this manuscript emphasizes on the fact that interaction with bystander cells of the tumor microenvironment can induce cancer cell heterogeneity. We used E-cad, ER, PR, Her2/neu, and EGFR1 simply as markers of heterogeneity progression, not because these proteins were more important but because their statuses of expression in MCF-7 cells were well known.
In agreement with the Reviewer’s suggestion, we are currently examining expressional changes, some of which may account for the morphologic and behavioral diversifications during successive co-culture. Such a study identifies differentially expressed proteins during heterogeneity progression. On the other hand, though we have previously reported our study on stemness-related proteins in association with MCF-7 malignant progression, as well as AR function in prostate cancers, whether these proteins are playing driver role in heterogeneity progression have yet to be determined. These studies, however, are beyond the scope of this manuscript.
In the revised manuscript, we added a new section, “4.5. Limitations of the study”, to address this issue. A new citation is added to remind readers of the limitation of this report.
- To deepen the background I strongly suggest the following manuscripts (doi:10.4252/wjsc.v11.i9.594; https://doi.org/10.3389/fendo.2018.0049).
Our response: We are indebted to the Reviewer for the constructive suggestion. In the revised manuscript, one of these review articles is cited to introduce the potential involvement of AR in breast cancer progression.
Reviewer 2 Report
The authors describe how the interactions between MCF-7 and HC can affect cancer cells metastatic potential and particularly CTC formation.
Of interest is the red fluorecensce methodology that better allows the investivation of cellular changes after interactions.
All experiments are in vitro and the clinical assumption are theoretical. Thereafter such consideration should be reported consequently and discussion should improve describing also the limitations of these experiments.
Author Response
- All experiments are in vitro and the clinical assumption are theoretical.
Our response: In the revised version, we discussed this issue by pointing out the fact that whether interaction with HCs in vivo may convert tumor cells with similar changes has not been corroborated.
- Thereafter such consideration should be reported consequently and discussion should improve describing also the limitation of these experiments.
Our response: We thank the Reviewer for the suggestion. In the revised manuscript, we added a section, “4.5. Limitations of the study”, to address two specific issues: 1) though this study established a reproducible experimental model, additional studies are needed to elucidate the mechanism of cell-cell interaction-induced heterogeneity progression; and 2) studies with additional cancer cell types are needed, and whether the model recapitulates in vivo heterogeneity progression of the disease remains to be investigated.
Round 2
Reviewer 1 Report
Authors addressed exaustively my comments